# MoLoRA: Composable Specialization via Per-Token Adapter Routing

**Shrey Shah** [1]   **Justin Wagle** [1]

## Abstract

Multi-adapter serving systems route entire sequences to a single adapter, forcing a choice when requests span multiple domains. This assumption fails in two important settings: (1) multimodal generation, where text and image tokens require different adapters within the same sequence, and (2) mixed-capability requests like "write code to solve this equation," which need expertise from multiple specialized adapters. We introduce *per-token routing*, which routes individual tokens to adapters based on either vocabulary structure (for multimodal models) or learned gating (for semantic specialization). Per-token routing is provably optimal for mixed-adapter requests: $N$ work for $N$ tokens, versus $K \cdot N$ for per-sequence systems that must replay one adapter per pass. Our key contribution is MoLoRA (Mixture of LoRA), which enables *composable specialization*: load multiple domain-specific adapters and let a learned router select the appropriate adapter per-token. We demonstrate that specialization dramatically beats scale: MoLoRA enables Qwen3-1.7B to exceed Qwen3-8B across four reasoning benchmarks while being $4.7\times$ smaller. This enables modular expertise at inference time: train focused LoRAs independently, combine them without retraining, and add new capabilities by simply loading new adapters.

## 1. Introduction

Low-rank adaptation (LoRA) (Hu et al., 2022) enables efficient fine-tuning of large language models by learning low-rank updates to pretrained weights. Multi-adapter serving systems such as S-LoRA (Sheng et al., 2024) and Punica (Chen et al., 2024a) extend this to serve multiple

[1]Microsoft. Correspondence to: Shrey Shah <shreyshah@microsoft.com>, Justin Wagle <justi-wag@microsoft.com>.

*Proceedings of the 43$^{rd}$ International Conference on Machine Learning*, Seoul, South Korea. PMLR 306, 2026. Copyright 2026 by the author(s).

adapters concurrently, enabling a single base model deployment to serve many fine-tuned variants.

However, existing systems share a fundamental assumption: each request routes to exactly one adapter. Formally, given a batch of $N$ tokens from $B$ sequences, these systems compute the routing function $j : [B] \to [K]$ mapping sequences to adapters, then apply:

$$h_i = x_i W + x_i A_{j(s_i)} B_{j(s_i)} \tag{1}$$

where $s_i$ denotes the sequence containing token $i$. This *per-sequence* routing causes two problems:

**Multimodal Efficiency.** Frontier models like Gemini (Gemini Team, 2025) generate interleaved text and images in a single response—an illustrated recipe where text instructions alternate with generated images. With per-sequence routing, all tokens must use the same adapter, even though text and image tokens should use modality-specialized adapters. This forces either suboptimal adapter selection or expensive sequence splitting ($K$ forward passes for $K$ modalities).

**Mixed-Capability Quality.** Consider a request like "write Python code to solve this differential equation." A code adapter excels at syntax but not calculus; a math adapter handles equations but not programming idioms. With per-sequence routing, we must choose one and accept suboptimal quality on the other capability. No single adapter can match multiple specialists.

We introduce *per-token routing*, which generalizes Equation 1 to route individual tokens:

$$h_i = x_i W + x_i A_{r(i)} B_{r(i)} \tag{2}$$

where $r : [N] \to [C]$ routes each token to one of $C$ computational targets. Per-token routing solves both problems: it reduces $K$ forward passes to 1 (Problem 1), and enables different tokens to use different specialized adapters within the same sequence (Problem 2). The routing function $r$ can be deterministic (based on vocabulary structure, for multimodal models) or learned (for semantic specialization).

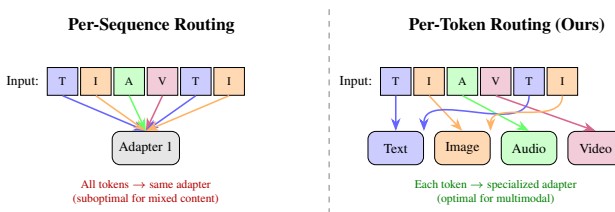

*Figure 1.* Per-sequence routing (left) sends all tokens in a sequence to the same adapter, even when tokens have different modalities (T=text, I=image, A=audio, V=video). Per-token routing (right) routes each token to its modality-specialized adapter within a single forward pass.

**Contributions.** We make three contributions:

1. **Per-token routing framework** (§3). We formalize per-token routing and prove it is computationally optimal for mixed-route requests under one-adapter-per-pass execution: work $N$ for $N$ tokens versus $K \cdot N$ for per-sequence routing when one logical request requires $K$ adapter types.

2. **MoLoRA: Composable specialization** (§4). We introduce MoLoRA (Mixture of LoRA), which extends per-token routing with learned gating. MoLoRA enables *composable specialization*: load multiple domain-specific adapters and let a learned router select per-token. We demonstrate that **specialization beats scale**: Qwen3-1.7B + MoLoRA exceeds Qwen3-8B (4.7× larger) on all four reasoning benchmarks—GSM8K (+14%), MATH (+8%), BBH (+2.5%), GPQA (+2.1%). A single 1.7B union-LoRA trained on the same data underperforms MoLoRA on every benchmark, isolating the gain from routing rather than dataset union.

3. **Systems and empirical validation** (§5–7). A hot-set memory architecture enables CUDA graph capture, reducing P99 latency by 67×. Per-token routing achieves $K\times$ improvement for $K$-modality workloads: 4.1× from pass reduction, compounding to 5.5× with systems optimizations.

This paper separates three layers of the design: the routing granularity (per-token rather than per-sequence), the routing policy (deterministic vocabulary routing or learned MoLoRA gating), and the execution substrate (hot-set memory with grouped dispatch and CUDA graph capture). This separation lets us isolate quality gains from composable specialization and latency gains from pass reduction and graph-captured execution.

## 2. Background and Related Work

**LoRA and Multi-Adapter Serving.** LoRA (Hu et al., 2022) fine-tunes pretrained weights via low-rank updates

$\Delta W = AB$ where $A \in \mathbb{R}^{d \times r}$, $B \in \mathbb{R}^{r \times d}$, adding only $2dr$ parameters per layer. Multi-adapter systems like S-LoRA (Sheng et al., 2024) and Punica (Chen et al., 2024a) serve multiple adapters through unified paging and specialized kernels, but assume per-sequence routing: $j : [B] \to [K]$ maps sequences to adapters, precluding per-token differentiation. Work on LoRA composition (Wu et al., 2024; Zhong et al., 2024) and per-token selection (Belofsky, 2023; Chen et al., 2024b) focuses on training or generalization; we target serving efficiency with systems optimizations absent from prior approaches.

**MoE and Infrastructure Unification.** MoE architectures (Shazeer et al., 2017; Fedus et al., 2022) route tokens to experts via learned gating. Per-token routing brings adapter serving onto the same systems abstraction as MoE dispatch: both use histogram construction, pointer-based scatter-gather, and grouped computation. We adopt adaptive tiling from MoE systems (Gale et al., 2023), and the dispatch kernel is target-agnostic—optimizations transfer bidirectionally.

**Multimodal Vocabularies.** Unified multimodal architectures like Chameleon (Chameleon Team, 2024) encode modality in token indices: text tokens occupy one vocabulary range, image tokens another. We exploit this for deterministic per-token routing via vocabulary boundary comparisons.

## 3. Per-Token Routing Framework

We develop a framework for per-token adapter routing, establishing its relationship to per-sequence routing and mixture-of-experts.

### 3.1. From Sequences to Tokens

Let $\mathcal{X} = \{x_1, \ldots, x_N\}$ denote a batch of $N$ tokens with associated vocabulary indices $v_i \in [V]$ and sequence memberships $s_i \in [B]$. Per-sequence routing computes $r(i) = j(s_i)$ where $j : [B] \to [K]$ assigns adapters to sequences. This constrains all tokens in a sequence to use the same adapter.

Per-token routing removes this constraint by defining $r : [N] \to [C]$ directly on tokens. The key insight is that multimodal vocabularies encode routing information:

**Definition 3.1** (Vocabulary Routing). Given vocabulary breaks $\mathbf{b} = (b_0, b_1, \ldots, b_M)$ with $b_0 = 0$ and $b_M = V$, the vocabulary routing function is:

$$\mathcal{R}_{\text{vocab}}(v) = m \iff v \in [b_{m-1}, b_m) \qquad (3)$$

For a multimodal model with $M$ modalities (Chameleon Team, 2024), tokens route to adapters based on their vocabulary range: text tokens to a text adapter, image tokens to

an image adapter, audio to audio, video to video, and so forth. The routing decision requires only $M - 1$ integer comparisons.

**Assumption.** This formulation assumes modalities occupy contiguous vocabulary ranges, as in unified multimodal tokenizers. Non-contiguous assignments require an $O(V)$ lookup table; we focus on the contiguous case which covers Chameleon-family models and similar architectures.

### 3.2. Compositional Routing

Per-token routing enables routing over *product spaces*, providing exponential expressiveness from independently trained components.

**Definition 3.2** (Compositional Routing). Given adapter set $\mathcal{A}$ and modality set $\mathcal{M}$, compositional routing defines targets $\mathcal{C} = \mathcal{A} \times \mathcal{M}$ with routing function:

$$r(i) = (a_i, m_i) \in \mathcal{A} \times \mathcal{M} \qquad (4)$$

where $a_i$ is determined by request metadata (e.g., customer identity) and $m_i = \mathcal{R}_{\text{vocab}}(v_i)$ is determined by vocabulary structure.

The composite target $c = a \cdot |\mathcal{M}| + m$ indexes into a weight tensor of shape $(|\mathcal{A}|, |\mathcal{M}|, d, r)$, providing $|\mathcal{A}| \cdot |\mathcal{M}|$ distinct computational paths with a single dispatch operation. The key advantage is *unified serving*: rather than deploying $|\mathcal{A}| \cdot |\mathcal{M}|$ separate model instances, a single deployment handles all combinations through compositional indexing.

**Example 3.3.** With 64 customer adapters and 4 modalities, compositional routing provides 256 computational paths from a single $(64, 4, d, r)$ weight tensor, served through unified infrastructure rather than 256 separate deployments.

The framework extends naturally to additional factors. Token-level specialization $\mathcal{T}$ (e.g., routing special tokens differently) yields product space $\mathcal{A} \times \mathcal{M} \times \mathcal{T}$.

### 3.3. Unified Dispatch Infrastructure

The dispatch kernel that routes tokens to targets in a LoRA setting uses the same target-grouped structure as routing tokens to MoE experts. Given routing decisions $r \in [C]^N$, the kernel:

1. Constructs a histogram $h \in \mathbb{Z}^C$ counting tokens per target

2. Allocates positions via atomic increment: pos $\leftarrow$ atomicAdd($h[r[t]], 1$)

3. Emits pointer arrays $\mathbf{xs}[c, \text{pos}], \mathbf{ys}[c, \text{pos}]$ for scatter-gather

This three-step pattern is target-agnostic: the kernel operates on abstract routing decisions without knowledge of whether targets represent experts, adapters, or compositional combinations thereof. The histogram enables adaptive tiling in subsequent compute kernels—targets with few tokens use smaller tiles while targets with many tokens use larger tiles for better memory bandwidth utilization (Appendix D).

For compositional routing, the kernel computes composite targets $c = a \cdot |\mathcal{M}| + m$ by combining adapter indices from request metadata with modality indices from vocabulary structure. The complete algorithm appears in Appendix D.

### 3.4. Theoretical Analysis

**Proposition 3.4** (Routing Complexity). *Vocabulary routing achieves $\mathcal{O}(1)$ per-token routing cost for fixed $M$, compared to $\mathcal{O}(E \cdot d)$ for learned MoE gating over $E$ experts with hidden dimension $d$.*

Vocabulary routing requires $M - 1$ integer comparisons, which are unrolled at compile time for typical $M \leq 4$. Learned gating computes softmax($Wx + b$) where $W \in \mathbb{R}^{E \times d}$, requiring $E \cdot d$ multiplications. For $d = 4096$ and $E = 8$, this represents a $\sim 10,000 \times$ difference in routing overhead.

**Proposition 3.5** (Expressiveness). *When the optimal routing function $\mathcal{R}^*$ satisfies $\mathcal{R}^*(x) = m$ iff $v(x) \in [b_{m-1}, b_m)$, vocabulary routing achieves $\mathcal{R}_{vocab} = \mathcal{R}^*$.*

When vocabulary structure determines optimal routing— as in multimodal models where modality determines the appropriate adapter—deterministic routing matches learned routing without parameter overhead.

### 3.5. Key Theoretical Results

We establish two results that distinguish per-token from per-sequence routing: computational optimality for mixed-route requests and a unification with sparse attention. In Theorem 3.6, $K$ denotes the number of distinct adapter targets required within a logical request, not the total adapter catalog size.

**Theorem 3.6** (Computational Optimality). *Consider a logical request with $N$ tokens that require $K$ distinct adapter targets. Under a per-sequence execution interface that applies at most one adapter to a sequence per pass, a correct implementation must either choose one adapter for all tokens (incorrect for $K > 1$) or replay/split the request into $K$ route-homogeneous passes. When each pass preserves full-sequence context, the adapter-layer work satisfies:*

$$W_{per\text{-}seq} \geq K \cdot N \cdot c_{pass} \quad (per\text{-}sequence) \qquad (5)$$

$$W_{per\text{-}tok} = N \cdot c_{pass} \quad (per\text{-}token) \qquad (6)$$

*where $c_{pass}$ is the cost of processing one token through*

*one adapter. Per-token routing achieves the minimum one adapter application per token.*

The factor $K$ is therefore a property of mixed-route requests under one-adapter-per-pass execution. If all tokens use the same adapter, then $K = 1$ and the gap disappears.

We also establish an equivalence between per-token adapter routing and sparse attention with block-diagonal patterns (Appendix C), enabling optimizations to transfer between sparse attention and multi-adapter systems.

## 4. MoLoRA

We now present **MoLoRA (Mixture of LoRA)**, which extends per-token routing with learned gating to enable *composable specialization*—the ability to load multiple specialized LoRA adapters simultaneously and route tokens dynamically based on content.

### 4.1. Motivation

Traditional adapter serving requires choosing a single adapter per request: a math LoRA for mathematical reasoning, a code LoRA for programming, a creative LoRA for writing. However, real-world requests often require multiple capabilities. A request like "write Python code to solve this differential equation" needs both mathematical and programming expertise. Per-sequence routing forces a choice, accepting suboptimal quality on one capability.

Per-token routing with learned gating eliminates this trade-off. By loading multiple specialized adapters and routing per-token, a single serving endpoint achieves the quality benefits of all fine-tunes combined:

- **Single-domain tasks**: The router selects the specialized adapter, matching single-adapter quality
- **Mixed-capability tasks**: The router selects different adapters for different tokens, combining expertise within a single sequence

This enables *modular expertise at inference time*: train focused LoRAs independently, combine them without retraining, and add new capabilities by loading new adapters.

### 4.2. When Vocabulary Routing Is Insufficient

Deterministic vocabulary routing requires modality information encoded in token indices. We identify four scenarios where this assumption fails, motivating learned routing.

**Scenario 1: Encoder-Based Multimodal Models.** Models like LLaVA (Liu et al., 2023), Flamingo (Alayrac et al., 2022), and Qwen-VL (Bai et al., 2023) process images through separate encoders (e.g., CLIP (Radford et al., 2021)) before projecting into the LLM's embedding space. The

resulting "image tokens" occupy the *same vocabulary range* as text tokens. Unlike Chameleon's disjoint ranges, these models provide no vocabulary-level signal distinguishing modalities.

**Scenario 2: Semantic Specialization.** Consider adapters specialized for code, mathematical reasoning, creative writing, and technical documentation. These domains share the same vocabulary—the word "function" appears in all four contexts with identical token IDs. Yet optimal adaptation differs: code adapters should emphasize syntax patterns, math adapters logical structure. Vocabulary-based routing cannot distinguish these cases.

**Scenario 3: Sub-Modality Granularity.** Even when vocabulary encodes modality, finer-grained specialization may be valuable. Within image tokens, photographs, diagrams, and charts may benefit from different adapters. Vocabulary routing provides only coarse modality-level grouping; learned routing enables arbitrary granularity.

**Scenario 4: Compositional Multi-Attribute Routing.** Production settings often require routing along multiple dimensions: modality × domain × task. Vocabulary routing handles only one dimension. Learned routing naturally extends to product spaces by predicting multi-dimensional routing targets.

### 4.3. Router Architecture

Given input $x \in \mathbb{R}^{B \times L \times d}$, where $B$ is batch size, $L$ is sequence length, $d$ is hidden size, and $K$ is the number of loaded adapters, the router $g_\theta : \mathbb{R}^d \to \mathbb{R}^K$ produces adapter logits per token. We apply top-$k$ selection followed by softmax:

$$w_i = \text{softmax}(\text{TopK}(g_\theta(x_i), k)) \tag{7}$$

$$\Delta h_i = \sum_{j \in \text{TopK}} w_{i,j} \cdot x_i A^{(j)} B^{(j)} \tag{8}$$

The router is a 2-layer MLP with hidden dimension 64 and GELU activation:

$$g_\theta(x) = W_2 \cdot \text{GELU}(W_1 x + b_1) + b_2 \tag{9}$$

This adds minimal parameters ($64d + 64K$) while enabling input-dependent adapter selection. The top-$k$ value controls the composition–cost operating point: larger $k$ permits smoother composition across adapters, while small $k$ keeps compute close to a single routed LoRA application. Following Switch Transformer (Fedus et al., 2022), we add an auxiliary load-balancing loss to encourage uniform adapter utilization.

**Load balancing.** The auxiliary loss discourages training-time router collapse. At inference time, routing does not require a hard MoE-style capacity constraint: skewed batches create larger grouped-GEMM bins, often improving arithmetic intensity, while histogram-guided grouping and adaptive tiling maintain efficiency for fragmented distributions.

### 4.4. Composable Specialization Results

We evaluate the central claim: MoLoRA enables composable specialization, where multiple domain-specific adapters combine at inference time to match specialized performance across all domains.

**Setup.** We use **Qwen3-1.7B** as the base model and train four specialized LoRA adapters (rank=32) using GRPO (Shao et al., 2024) on a filtered general-purpose reasoning corpus spanning different domains: math, logical reasoning, and scientific reasoning. A lightweight router (2-layer MLP) learns to classify tokens and select the appropriate adapter. We evaluate on the standard test splits of GSM8K, MATH, BBH, and GPQA, comparing against **Qwen3-8B** ($4.7\times$ larger) and a **1.7B union-LoRA** trained on the same union of reasoning data without routing.

**Results.** Figure 3 demonstrates that **MoLoRA beats scale across all benchmarks**:

- **vs Qwen3-8B:** MoLoRA wins on GSM8K (+14%), MATH (+8%), BBH (+2.5%), GPQA (+2.1%)
- **vs union-LoRA:** MoLoRA improves over a single adapter trained on the same union data by +3.0% GSM8K, +14.5% MATH, +7.0% BBH, and +2.1% GPQA
- **Composability:** A single model with four adapters handles all reasoning domains

The 1.7B model with MoLoRA achieves this while being **$4.7\times$ smaller** than the 8B model. The union-LoRA control shows the gain is not just data union; MoLoRA avoids monolithic-adapter negative transfer.

**Learned Routing.** MoLoRA's router learns to distinguish between reasoning domains—grade-school math, competition math, logical reasoning, and scientific reasoning—and selects appropriate adapters per-token automatically. The learned assignments recover the intended domain-specific structure, enabling a single deployment to handle diverse reasoning tasks without manual adapter selection.

**Implications.** Rather than training larger models, practitioners can achieve better results by training small, focused adapters and combining them via MoLoRA. New capabilities require only training and loading a new LoRA—existing adapters need not be retrained.

### 4.5. Production-Grade Infrastructure

Composable specialization is only practical if it can be deployed efficiently at scale. A key design property of our system is that the same dispatch and compute kernels support both deterministic and learned routing (Table 1).

*Table 1.* Infrastructure reuse between vocabulary routing and MoLoRA. Only the routing decision differs; post-routing computation is identical.

| Component | Vocabulary Routing | MoLoRA |
|---|---|---|
| Routing decision | $\mathcal{R}_{\text{vocab}}(v)$ | $\text{TopK}(g_\theta(x))$ |
| Routing cost | $\mathcal{O}(1)$ | $\mathcal{O}(64d + 64K)$ |
| *Identical Infrastructure (Reused)* | | |
| Histogram computation | ✓ | ✓ |
| Pointer arrays | ✓ | ✓ |
| Grouped GEMM dispatch | ✓ | ✓ |
| Adaptive tiling | ✓ | ✓ |
| CUDA graph capture | ✓ | ✓ |

This means MoLoRA inherits all the systems benefits developed in §5–6: hot-set memory for CUDA graph capture, per-token dispatch for $K\times$ pass reduction, and adaptive tiling for grouped computation. MoLoRA replaces vocabulary-based routing with a learned gating function while reusing identical post-routing infrastructure.

This validates the MoE/multi-adapter correspondence: once tokens are assigned to targets (via any mechanism), the grouped computation is identical.

## 5. System Architecture

We now describe the execution substrate used after routes have been assigned. The hot-set memory model is independent of the routing policy: vocabulary routing, MoLoRA, and even per-sequence routing can benefit from fixed GPU-resident adapter slots. In our system, it complements per-token routing by making grouped execution predictable and graph-capturable.

### 5.1. Dynamic Paging Bottlenecks

S-LoRA and Punica employ CPU-GPU paging to support adapter catalogs exceeding GPU memory. When a requested adapter is not GPU-resident, the system evicts an LRU adapter and copies the requested adapter from CPU memory. This design introduces variable latency on the critical path and prevents CUDA graph capture due to data-dependent control flow.

### 5.2. Static Hot-Set Memory Model

We propose a *hot-set* architecture that pre-allocates $S$ adapter slots on GPU:

**Definition 5.1** (Hot-Set Layout). Given $S$ adapter slots, $M$

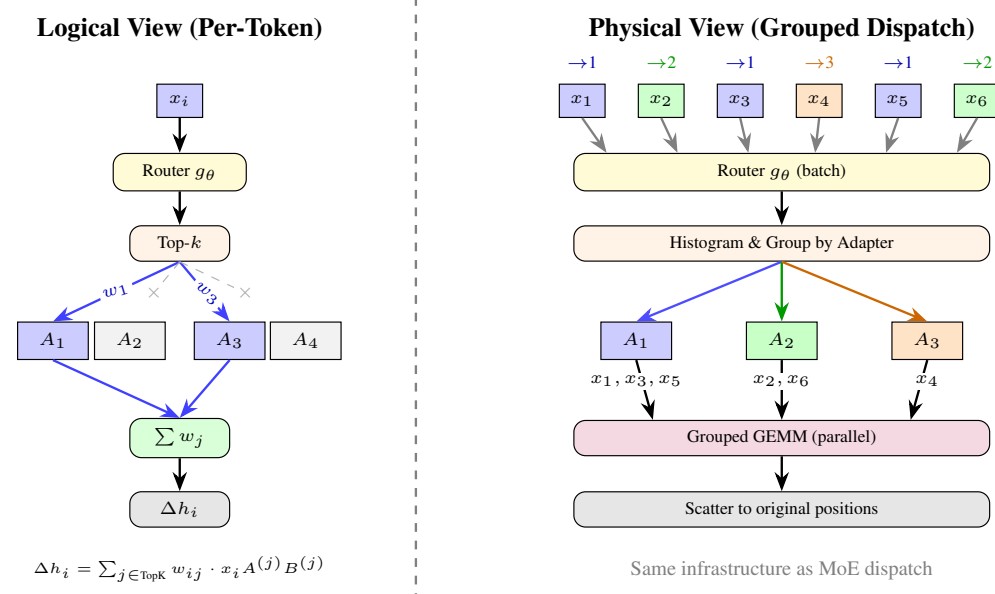

Figure 2. MoLoRA: logical vs. physical execution. **Left:** Per-token view—a router selects top-$k$ adapters (here $k$=2), which are combined via weighted sum. **Right:** Batched execution—tokens are grouped by their selected adapter, enabling parallel grouped GEMM using identical infrastructure to MoE systems.

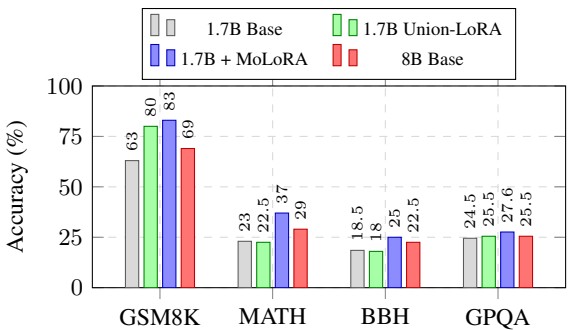

Figure 3. MoLoRA enables small models to exceed larger ones. Qwen3-1.7B with four specialized LoRA adapters and learned routing (blue) exceeds Qwen3-8B (red) and a single union-LoRA trained on the same data (green), while being $4.7\times$ smaller than 8B.

modalities, model dimension $d$, and LoRA rank $r$:

$$\mathbf{A}_{\text{hot}} \in \mathbb{R}^{S \times M \times d \times r} \tag{10}$$

$$\mathbf{B}_{\text{hot}} \in \mathbb{R}^{S \times M \times r \times d} \tag{11}$$

A slot table $\sigma : [S] \to \mathcal{A}$ maps slots to adapters, maintained asynchronously.

The static memory layout provides three properties: (1) fixed addresses enabling CUDA graph capture, (2) no paging latency on the critical path, and (3) predictable memory consumption. The hot-set represents the active working set rather than the full adapter catalog: large deployments can keep frequently used adapters resident and graph-captured, while serving cold adapters through a paging path. This hybrid deployment preserves predictable latency for the

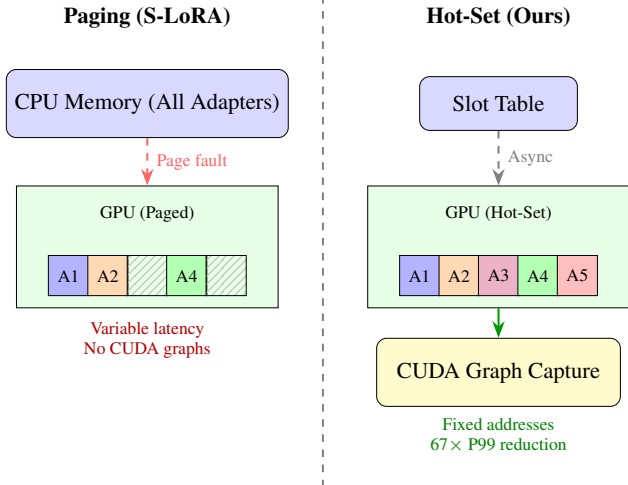

Figure 4. System architecture comparison. Paging (left) copies adapters on-demand from CPU, causing variable latency and preventing CUDA graph capture. Our hot-set architecture (right) pre-allocates GPU-resident adapters with fixed addresses, enabling graph capture and predictable latency.

common case without requiring the entire catalog to fit in GPU memory.

### 5.3. CUDA Graph Integration

With static hot-set memory, the forward pass has fixed memory addresses and deterministic control flow, enabling CUDA graph capture. Graph replay eliminates per-kernel launch overhead (5–10$\mu$s per kernel), driver scheduling

latency, and Python/C++ boundary crossings. For a multi-adapter forward pass with 10+ kernel launches, graph capture reduces launch overhead from 50–100$\mu$s to $<10\mu$s.

### 5.4. Architectural Decomposition

To understand where improvements originate, we systematically isolate each factor.

**Summary.** The hot-set architecture provides two benefits: (1) fixed memory addresses enable CUDA graph capture, yielding 67× P99 improvement (§7), and (2) direct indexing provides 1.14× improvement over indirect indexing (Table 7). Combined with per-token routing (35× average, §7) and kernel optimizations (1.3–2.7× over S-LoRA at production batch sizes), these architectural choices enable sub-millisecond latencies for multi-adapter serving.

## 6. Kernel Design

Post-routing computation applies the LoRA transformation to tokens grouped by target.

### 6.1. Kernel Architecture

Our implementation uses tensor-core HMMA operations with large tiles, multi-stage pipelining, and fused post-operations. Critically, we leverage CUDA graph capture to eliminate kernel launch overhead, achieving near-constant latency regardless of batch size. S-LoRA and Punica use scalar FMA operations (BGMV kernel) optimized for the memory-bound regime (Appendix B).

Our tensor-core implementation with CUDA graph capture targets the compute-bound regime where batch size provides sufficient arithmetic intensity for tensor cores to deliver higher throughput. The combination of tensor cores and graph capture is key: tensor cores provide raw compute throughput, while graph capture eliminates the Python/CUDA launch overhead that would otherwise dominate at small batch sizes.

### 6.2. Performance Characterization

Figure 5 shows the transition point at batch size ≈160. Our tensor-core implementation with CUDA graph capture achieves near-constant latency (∼0.016ms) across batch sizes, while S-LoRA's BGMV kernel scales linearly. At batch 512, our kernel is 1.98× faster; at batch 1024, 2.69× faster. Modern serving systems using continuous batching (Yu et al., 2022; Agrawal et al., 2024) aggregate tokens from multiple concurrent requests. Under loads targeting high GPU utilization, batch sizes of 256–2048 tokens are typical, where our kernel provides 1.3–2.7× speedup.

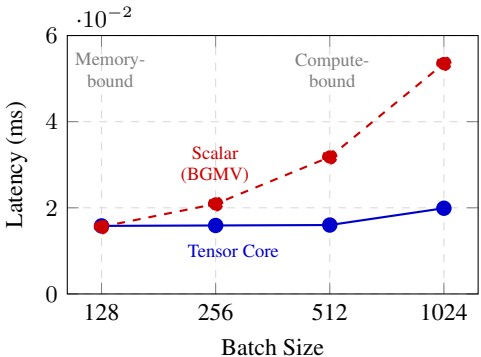

*Figure 5.* Kernel performance comparison. Scalar implementations (S-LoRA's BGMV) excel in the memory-bound regime (batch $<160$), while our tensor-core kernel with CUDA graph capture dominates in the compute-bound regime. The transition occurs at batch size ≈160, with our kernel achieving 1.98× speedup at batch 512.

*Table 2.* **Per-token routing reduces $K$ forward passes to 1.** Per-sequence routing requires $K$ passes for $K$-modality workloads; per-token routing requires exactly one. Configuration: $K$=4 modalities, 2048 tokens, $d$=4096, $r$=64.

| Routing | Passes | Latency | Latency/Pass |
|---|---|---|---|
| Per-sequence ($K$=4) | 4 | 5.88ms | 1.47ms |
| Per-token | 1 | 1.43ms | 1.43ms |
| **Speedup** | **4×** | **4.1×** | — |

## 7. Experimental Evaluation

We evaluate MoLoRA on multimodal throughput, latency predictability, and kernel performance. Configuration details appear in Appendix A. Our central result is that **per-token routing reduces $K$ passes to 1 for $K$-modality workloads, yielding $K$× improvement**. Additional gains from hot-set memory and CUDA graphs compound this to 5.5× in controlled settings, with workload-dependent variation from 5.8× to 112×.

### 7.1. Fundamental Speedup from Pass Reduction

The primary advantage of per-token routing is reducing the number of forward passes. With $K$ modalities interleaved within sequences, per-sequence routing requires $K$ separate passes (one per modality), while per-token routing requires exactly 1. For homogeneous requests, $K = 1$ and both approaches execute one adapter pass; the benefit appears when a logical request contains multiple route targets.

Table 2 shows this directly: per-sequence routing takes 5.88ms (4 passes × 1.47ms each), while per-token routing takes 1.43ms (1 pass). The 4.1× speedup matches the theoretical $K$× prediction, confirming that pass reduction is the dominant source of improvement.

*Table 3.* Incremental improvements from each architectural choice. Same configuration as Table 2.

| Configuration | Latency | vs Baseline | Incremental | Source |
|---|---|---|---|---|
| Per-seq + Paging | 7.48ms | 1.0× | — | Baseline |
| Per-seq + Hot-set | 5.88ms | 1.3× | 1.3× | Paging eliminated |
| Per-token + Hot-set | 1.43ms | 5.2× | 4.1× | Passes: $K \rightarrow 1$ |
| Per-token + Graph | 1.36ms | 5.5× | 1.05× | Launch overhead |

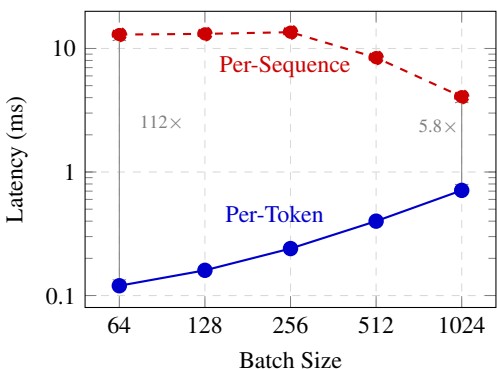

*Figure 6.* Latency comparison across batch sizes. Per-token routing (blue) maintains sub-millisecond latency while per-sequence routing (red) requires sequence splitting, incurring 5.8–112× higher latency. The gap narrows at larger batches as per-sequence overhead amortizes.

## 7.2. Ablations

To understand where gains originate, we incrementally add each optimization to a baseline of per-sequence routing with paging (S-LoRA style).

Table 3 shows the breakdown: (1) hot-set memory eliminates paging for 1.3×; (2) per-token routing reduces passes for 4.1×; (3) CUDA graph capture reduces launch overhead for 1.05×. Pass reduction is the dominant gain and represents the core algorithmic contribution; hot-set memory and CUDA graphs are systems optimizations that compound this benefit.

## 7.3. Workload-Dependent Scaling

The 4.1× improvement in Table 2 is for a specific configuration. Speedup varies with adapter diversity and batch size. On Qwen3-4B with 4 LoRA adapters, per-token routing achieves 4.1–4.3× speedup with diverse adapters, and parity (1.0×) when all sequences use the same adapter—showing that per-token routing matches or improves upon per-sequence routing across scenarios (Appendix E.2).

**Modality Distribution.** Speedup ranges from 7.5× (separated modalities, best case for per-sequence) to 52.7× (interleaved, worst case for per-sequence), averaging 28.7× across distributions (Appendix E.3). Per-token routing maintains constant latency regardless of token arrangement.

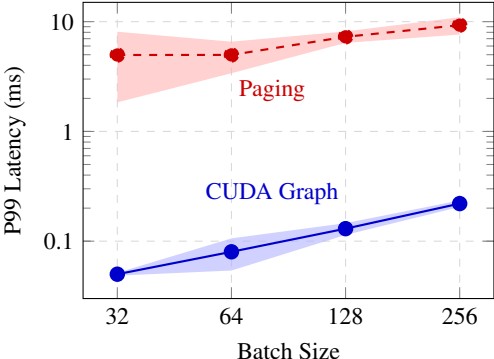

*Figure 7.* P99 latency with variance bands derived from coefficient of variation (CV). CUDA graph capture (blue) achieves 42–108× lower latency than paging (red). Shaded regions show ±1 std.

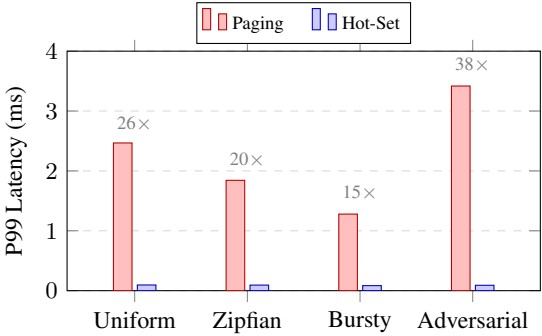

*Figure 8.* P99 latency under different access patterns. Hot-set latency remains stable (∼0.09ms) regardless of workload, while paging varies 3× between best-case (bursty) and worst-case (adversarial).

**Batch Size Scaling.** Speedup is highest at small batch sizes (112× at batch 64) because per-sequence routing overhead dominates. At larger batches, per-sequence routing amortizes overhead better, reducing the gap to 5.8× (Figure 6).

## 7.4. Latency and Variance

Figure 7 shows CUDA graph capture reduces P99 latency by 67× on average compared to dynamic paging. To isolate contributions: hot-set memory without graph capture (eager execution) achieves 0.08–0.44ms, while graph capture reduces this to 0.05–0.22ms—a 1–2× additional improvement. Paging elimination provides the largest latency drop; graph capture further reduces latency and, critically, stabilizes variance (see Appendix E.4).

## 7.5. Workload Robustness

Hot-set latency remains stable (0.084–0.095ms) regardless of access pattern, while paging degrades from 1.28ms (bursty) to 3.42ms (adversarial)—a 15–38× gap demonstrating predictable performance independent of workload

*Table 4.* End-to-end transformer latency. Per-sequence routing runs $K$ forward passes for $K$-modality workloads. Speedup is $\sim K\times$.

| Workload | Config | $K$ | Per-Token | Per-Seq | Speedup |
|---|---|---|---|---|---|
| Synthetic | $d=4096, L=4$ | 4 | 121.6ms | 488.0ms | 4.0$\times$ |
| Chameleon-style | $d=2048, L=4$ | 2 | 3.14ms | 5.18ms | 1.65$\times$ |

characteristics (Figure 8).

**Summary.** At production batch sizes (256–1024), speedup converges to $\sim K\times$ from pass reduction. The extreme values ($112\times$ at batch 64, $5.8\times$ at batch 1024) bound the range.

### 7.6. End-to-End Validation

To validate that kernel-level improvements translate to full models, we benchmark complete transformers with per-token LoRA routing.

Table 4 shows per-token routing achieves $K\times$ speedup end-to-end. The Chameleon-style benchmark (Chameleon Team, 2024) (interleaved text and image tokens) achieves $1.65\times$ for $K = 2$ modalities (vs. theoretical $2\times$); the gap arises because base transformer computation is shared. With $K = 4$ modalities, speedup approaches $4\times$. This validates that kernel-level improvements transfer to production models.

## 8. Conclusion

We introduced per-token routing for multi-adapter serving, addressing both the efficiency overhead of processing interleaved multimodal content and the quality compromise of forcing a single-adapter choice on mixed-capability requests. Per-token routing reduces $K$ forward passes to 1 for mixed-route workloads and unifies adapter dispatch with MoE-style grouped computation—histogram construction, pointer-based scatter-gather, and grouped computation become target-agnostic (and thus inherit MoE systems optimizations). Combined with a hot-set architecture enabling CUDA graph capture, our system achieves $67\times$ P99 latency reduction and sub-millisecond latencies. MoLoRA extends per-token routing with learned gating so mixed-capability requests can leverage multiple specialists within a single sequence. We demonstrate that specialization beats scale: Qwen3-1.7B with MoLoRA exceeds Qwen3-8B ($4.7\times$ larger) and a single union-LoRA control across four reasoning benchmarks. This enables modular expertise at inference time—new capabilities require only training and loading a new LoRA, without retraining existing adapters. As multimodal and multi-capability models become prevalent, per-token routing with composable specialization provides a principled foundation for efficient, high-quality multi-adapter serving.

## Impact Statement

This paper presents work whose goal is to advance the field of Machine Learning, specifically in efficient multi-adapter serving for large language models. The techniques we propose, per-token routing and composable specialization, reduce computational costs and enable smaller models to match larger ones, which has positive environmental implications through reduced energy consumption. There are many potential societal consequences of our work, none that we feel must be specifically highlighted here.

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

## A. Experimental Setup

**Hardware.**   Experiments were conducted on an NVIDIA H100 GPU with CUDA 13.0 and PyTorch 2.9.0.

**Model Configuration.**   We use model dimension $d = 4096$, LoRA rank $r = 64$, and 4 modalities.

**Baselines.**

- **Per-sequence routing**: S-LoRA/Punica-style implementation with specialized kernels

- **Dynamic paging**: LRU eviction with CPU-GPU transfers on cache miss

**Measurement.**   All latency measurements report the median of 1000 iterations after 100 warmup iterations.

## B. Kernel Implementation Details

### B.1. Adaptive Tiling

Post-routing computation uses adaptive tile sizes based on histogram counts, since different group sizes benefit from different tile configurations.

**Tile Selection Strategy.**   Given histogram counts $h[k]$ for each target $k$, we select tile sizes as:

$$\text{BLOCK\_M} = \begin{cases} 16 & h[k] < 64 \\ 32 & 64 \leq h[k] < 256 \\ 64 & h[k] \geq 256 \end{cases} \quad (12)$$

$$\text{BLOCK\_N} = \begin{cases} 32 & h[k] < 128 \\ 64 & h[k] \geq 128 \end{cases} \quad (13)$$

Smaller tiles reduce wasted computation when groups are small (avoiding padding overhead), while larger tiles improve memory bandwidth utilization through better cache locality when groups are large.

**Performance Impact.**   Table 5 shows the benefit of adaptive tiling across different token distributions.

*Table 5.* Adaptive vs. fixed tiling. Adaptive selection provides up to 1.4× improvement for skewed distributions.

| Distribution | Fixed (64×64) | Adaptive | Speedup |
|---|---|---|---|
| Uniform (25% each) | 0.412ms | 0.398ms | 1.04× |
| Skewed (80/10/5/5) | 0.456ms | 0.389ms | 1.17× |
| Extreme (95/2/2/1) | 0.521ms | 0.372ms | 1.40× |

The benefit is largest for skewed distributions where some adapters have very few tokens. Fixed 64×64 tiles waste computation on small groups; adaptive 16×32 tiles reduce this overhead.

**Implementation.**   We implement adaptive tiling in Triton by dispatching to different kernel configurations based on histogram counts. Each configuration is pre-compiled; the histogram determines which to invoke. This adds minimal overhead ($<1\mu s$ for histogram analysis) while providing significant benefits for non-uniform workloads.

### B.2. Kernel Performance Analysis

Scalar kernels (S-LoRA's BGMV) achieve superior small-batch performance through CUDA-specific optimizations: `cuda::memcpy_async` for pipelined memory operations, `__shfl_down_sync` for warp-level reductions, and hand-tuned thread configurations. These optimizations are most effective in the memory-bound regime. Our tensor-core implementation with CUDA graph capture eliminates kernel launch overhead, achieving near-constant latency across batch sizes. At batch sizes $\geq 160$, our kernel outperforms BGMV.

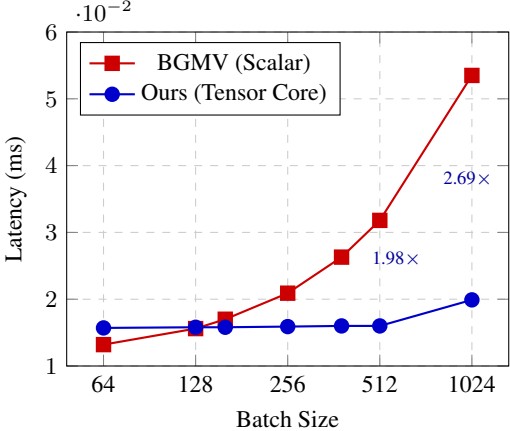

*Figure 9.* Kernel comparison: Tensor Core (with CUDA graph) vs BGMV scalar kernel. Our kernel achieves near-constant latency ($\sim 0.016$ms) while BGMV scales linearly. The transition occurs near batch 160; at production sizes (512–1024), we achieve 1.98–2.69× speedup. Benchmarked on H100 NVL with $d$=4096, $r$=64, 8 adapters.

Figure 9 shows the kernel comparison. The transition occurs at batch 160, where our tensor-core kernel begins to outperform BGMV. At production batch sizes (512–1024), we achieve 1.98–2.69× speedup. CUDA graph capture eliminates Python and kernel launch overhead, making our kernel's latency nearly constant ($\sim 0.016$ms) while BGMV scales linearly with batch size.

## C. Sparse Attention Equivalence

**Theorem C.1** (Sparse Attention Equivalence)**.** *Per-token adapter routing with $K$ adapters and vocabulary routing function $\mathcal{R}_{vocab}$ is equivalent to sparse attention with a block-diagonal attention pattern, where the vocabulary partition-*

*ing* determines the block structure.

*Proof.* Define attention weights $\alpha_{ij} = \mathbf{1}[\mathcal{R}_{\text{vocab}}(v_i) = \mathcal{R}_{\text{vocab}}(v_j)]$. The resulting attention pattern is block-diagonal with blocks corresponding to modality groups. The adapter computation $x_i A_{m(i)} B_{m(i)}$ can be written as:

$$\sum_j \alpha_{ij} \cdot x_j A_{\mathcal{R}_{\text{vocab}}(v_j)} B_{\mathcal{R}_{\text{vocab}}(v_j)} \quad (14)$$

which is a sparse attention operation where each token attends only to tokens of the same modality, using modality-specific projections. □

This equivalence unifies per-token adapter routing with the broader sparse attention literature. Optimizations developed for sparse attention (e.g., block-sparse patterns, hardware-efficient implementations) directly apply to adapter routing, and vice versa.

# D. Dispatch Infrastructure Details

The unified dispatch infrastructure described in §3.3 enables code reuse between MoE and multi-adapter systems. We detail the correspondence and implementation.

## D.1. Compositional Gating

Algorithm 1 shows the compositional gating kernel that combines adapter indices from request metadata with modality indices from vocabulary structure.

---

**Algorithm 1** Compositional Gating Kernel

**Require:** Token features $X$, vocab indices $V$, adapter indices $A$, modality breaks $\mathbf{b}$
**Ensure:** Histogram $h \in \mathbb{Z}^{|\mathcal{A}| \times |\mathcal{M}|}$, row pointers $\mathbf{xs}, \mathbf{ys}$
1: $h \leftarrow \mathbf{0}_{|\mathcal{A}| \times |\mathcal{M}|}$
2: **for** token $t = 1, \dots, N$ **in parallel do**
3:    $a \leftarrow A[t]$ {Adapter from request metadata}
4:    $m \leftarrow \text{FindModality}(V[t], \mathbf{b})$ {Modality from vocab}
5:    $c \leftarrow a \cdot |\mathcal{M}| + m$ {Composite target}
6:    $\text{pos} \leftarrow \text{atomicAdd}(h[c], 1)$
7:    $\mathbf{xs}[c, \text{pos}] \leftarrow \&X[t]; \mathbf{ys}[c, \text{pos}] \leftarrow \&Y[t]$
8: **end for**

---

## D.2. MoE and Multi-Adapter Correspondence

Table 6 provides a complete mapping between MoE and multi-adapter serving concepts. Post-routing computation is identical—only the routing decision mechanism differs.

*Table 6.* Complete correspondence between MoE and multi-adapter serving.

| Concept | MoE | Multi-Adapter |
|---|---|---|
| Routing function | $g_\theta(x) = \text{softmax}(Wx + b)$ | $j : [B] \to [K]$ or $\mathcal{R}_{\text{vocab}}(v)$ |
| Routing cost | $\mathcal{O}(E \cdot d)$ (learned) | $\mathcal{O}(1)$ (deterministic) |
| Routing decision | Per-token, learned | Per-sequence or per-token |
| Targets | Experts $\{1, \dots, E\}$ | Adapters $\{1, \dots, K\}$ |
| Weight shape | $(E, d_{\text{in}}, d_{\text{out}})$ | $(K, d, r)$ for $A$; $(K, r, d)$ for $B$ |
| Capacity | Expert capacity $C$ | Adapter memory budget |
| Load balancing | Auxiliary losses | Auxiliary losses or request shaping |
| *Identical Infrastructure* | | |
| Histogram | $h[e] = |\{t : r(t) = e\}|$ | $h[k] = |\{t : r(t) = k\}|$ |
| Pointer arrays | $\mathbf{xs}[e], \mathbf{ys}[e]$ | $\mathbf{xs}[k], \mathbf{ys}[k]$ |
| Dispatch kernel | Target-agnostic | Target-agnostic |
| Compute kernel | Grouped GEMM | Grouped GEMM |

## D.3. Unified Dispatch Algorithm

Algorithm 2 shows the complete dispatch kernel, which is identical for MoE and multi-adapter serving.

---

**Algorithm 2** Unified Dispatch Kernel (Target-Agnostic)

**Require:** Routing decisions $r \in [C]^N$, input pointer $x_{\text{ptr}}$, output pointer $y_{\text{ptr}}$, strides
**Ensure:** Histogram $h \in \mathbb{Z}^C$, pointer arrays $\mathbf{xs}, \mathbf{ys}$
1: $h \leftarrow \mathbf{0}_C$
2: **for** token $t = 1, \dots, N$ **in parallel do**
3:    $c \leftarrow r[t]$ {Target index (expert or adapter)}
4:    $\text{pos} \leftarrow \text{atomicAdd}(h[c], 1)$ {Allocate position}
5:    $\mathbf{xs}[c, \text{pos}] \leftarrow x_{\text{ptr}} + t \cdot \text{stride}_x$ {Input pointer}
6:    $\mathbf{ys}[c, \text{pos}] \leftarrow y_{\text{ptr}} + t \cdot \text{stride}_y$ {Output pointer}
7: **end for**

---

The histogram $h$ is then used by the compute kernel for adaptive tiling: targets with few tokens use smaller tiles, while targets with many tokens use larger tiles for better throughput.

## D.4. Implications for System Design

The dispatch unification has several practical implications:

1. **Code reuse**: Optimizations developed for MoE dispatch (e.g., MegaBlocks (Gale et al., 2023)) apply directly to multi-adapter serving.

2. **Hybrid systems**: A single dispatch infrastructure can route some tokens to experts (learned routing) and others to adapters (deterministic routing).

3. **Compositional routing**: The product space $\mathcal{A} \times \mathcal{M}$ is handled by computing composite indices $c = a \cdot |\mathcal{M}| + m$ before dispatch.

# E. Additional Results

## E.1. Memory Layout Comparison

Comparing direct indexing (our approach) against indirect indexing (S-LoRA/Punica style) using identical operations, direct memory indexing provides $1.14\times$ average improvement across adapter counts.

Table 7. Memory layout comparison using identical operations.

| Adapters | Indirect | Direct | Speedup |
|---|---|---|---|
| 4 | 0.215ms | 0.196ms | $1.10\times$ |
| 8 | 0.228ms | 0.194ms | $1.17\times$ |
| 16 | 0.227ms | 0.201ms | $1.13\times$ |
| 32 | 0.232ms | 0.207ms | $1.12\times$ |
| 64 | 0.258ms | 0.218ms | $1.18\times$ |
| **Average** | | | **$1.14\times$** |

## E.2. Production Model Validation

To validate that improvements transfer to production models, we benchmark on Qwen3-4B (Qwen Team, 2025) with 4 LoRA adapters (rank 8) targeting attention projections. Table 8 shows that speedup scales with adapter diversity: $4.1$–$4.3\times$ with diverse adapters, and parity ($1.0\times$) when all sequences use the same adapter.

Table 8. Qwen3-4B benchmark comparing per-sequence (S-LoRA style) vs per-token routing. Speedup depends on adapter diversity.

| Scenario | Batch | Seq | Per-Seq | Per-Token | Speedup |
|---|---|---|---|---|---|
| Diverse adapters | 4 | 128 | 179.0ms | 41.5ms | $4.3\times$ |
| Diverse adapters | 8 | 128 | 178.7ms | 43.4ms | $4.1\times$ |
| Uniform (best S-LoRA) | 8 | 256 | 178.1ms | 79.8ms | $2.2\times$ |
| Single adapter | 8 | 256 | 80.3ms | 80.0ms | $1.0\times$ |

## E.3. Modality Distribution Analysis

Speedup varies depending on how modalities are distributed within sequences. When modalities are interleaved, per-sequence routing must split at boundaries, incurring maximum overhead. When modalities are separated (each sequence is single-modality), per-sequence routing can batch efficiently. Table 9 shows this variation.

Table 9. Speedup varies with modality distribution. "Interleaved" requires sequence splitting; "separated" allows efficient batching.

| Distribution | Per-Token | Per-Sequence | Speedup |
|---|---|---|---|
| Interleaved, text-heavy | 0.23ms | 5.94ms | $26.0\times$ |
| Interleaved, balanced | 0.24ms | 12.79ms | $52.7\times$ |
| Separated (control) | 0.24ms | 1.81ms | $7.5\times$ |
| **Average** | | | **$28.7\times$** |

## E.4. Latency Variance

CUDA graph capture not only reduces latency but also reduces variance. Table 10 compares coefficient of variation (CV = std/mean) across execution modes. Graph capture achieves consistently low CV (0.03–0.33), while eager execution and paging show higher variance (up to 0.68).

Table 10. Latency variance comparison (CV = std/mean).

| Batch | Eager CV | Graph CV | Paging CV |
|---|---|---|---|
| 32 | 0.52 | 0.03 | 0.63 |
| 64 | 0.68 | 0.33 | 0.32 |
| 128 | 0.34 | 0.12 | 0.12 |
| 256 | 0.01 | 0.07 | 0.18 |

# F. Additional MoLoRA Results

This appendix contains additional MoLoRA experimental results that support the main text findings.

## F.1. Load Balancing Details

Following Switch Transformer (Fedus et al., 2022), we add an auxiliary loss to encourage uniform adapter utilization:

$$\mathcal{L}_{\text{aux}} = K \cdot \sum_{j=1}^{K} f_j \cdot p_j \tag{15}$$

where $f_j$ is the fraction of tokens routed to adapter $j$ (based on top-1 selection) and $p_j$ is the mean routing probability for adapter $j$. This loss penalizes configurations where high-probability adapters also receive many tokens, encouraging balanced utilization. This regularizes training; inference remains capacity-free. A hotspot adapter creates a larger grouped-GEMM bin, which is correct by construction and often more compute-efficient. Adaptive tiling targets the complementary fragmented case with many small bins.

## F.2. Use Case Taxonomy

Table 11 summarizes scenarios where vocabulary routing suffices versus where MoLoRA's learned routing is required. MoLoRA generalizes vocabulary routing to the listed semantic, sub-modality, and multi-attribute settings.

Table 11. MoLoRA use case taxonomy. Vocabulary routing requires modality-encoding vocabulary structure; MoLoRA extends routing to semantic, sub-modality, and multi-attribute settings.

| Scenario | Vocab Routing | MoLoRA |
|---|---|---|
| Chameleon-style (disjoint vocab) | ✓ | ✓ |
| Encoder-based multimodal (LLaVA, Flamingo) | ✗ | ✓ |
| Semantic specialization (code/math/prose) | ✗ | ✓ |
| Sub-modality granularity (photo/diagram/chart) | ✗ | ✓ |
| Multi-attribute routing (modality × domain) | ✗ | ✓ |

## F.3. Synthetic Multimodal Task

We evaluate MoLoRA on a synthetic multimodal task where each of 3 modalities has a distinct optimal transformation. The task tests whether MoLoRA can learn modality-specialized routing without access to modality labels.

**Setup.** We use $d = 256$, rank $r = 16$, and compare four approaches: (1) **Single**: one adapter for all tokens; (2) **Fixed (Oracle)**: ground-truth modality labels determine routing; (3) **MoLoRA**: learned routing with 4 adapters, top-$k = 2$; (4) **MoLoRA-L**: 8 adapters, top-$k = 3$.

*Table 12.* MoLoRA training results on synthetic multimodal task. MoLoRA learns routing without labels, achieving 35% of oracle improvement.

| Model | Adapters | top-$k$ | Train Loss | Val Loss |
|---|---|---|---|---|
| Single Adapter | 1 | – | 3.008 | 3.435 |
| Fixed (Oracle) | 3 | 1 | 2.588 | 3.819 |
| MoLoRA | 4 | 2 | 2.861 | 3.555 |
| MoLoRA-L | 8 | 3 | 2.783 | 3.599 |

Table 12 shows that MoLoRA reduces training loss by 4.9% over single-adapter (2.861 vs 3.008), achieving 35% of the oracle improvement. The oracle (fixed routing with known labels) achieves 14% improvement, demonstrating the value of modality-specialized adapters. MoLoRA approaches this without access to labels.

## F.4. Emergent Modality Discovery

A key finding is that MoLoRA's router automatically discovers modality structure without supervision.

*Table 13.* MoLoRA discovers modality structure without supervision. ARI (Adjusted Rand Index) and NMI (Normalized Mutual Information) measure alignment between learned routing and true modality labels. Higher is better; 1.0 indicates perfect clustering.

| Epoch | ARI | NMI | Routing Entropy |
|---|---|---|---|
| 0 (random) | 0.27 | 0.37 | 1.35 |
| 20 | 0.72 | 0.86 | 0.09 |
| 99 (converged) | **0.71** | **0.84** | 0.18 |

Table 13 reveals a surprising finding: **MoLoRA's router automatically discovers modality structure despite never receiving modality labels during training**. Starting from random routing (ARI=0.27), the router converges to near-perfect modality clustering (ARI=0.71, NMI=0.84). The confusion matrix shows each adapter specializes to specific modalities:

| | A0 | A1 | A2 | A3 |
|---|---|---|---|---|
| Modality 0 | **1.00** | 0.00 | 0.00 | 0.00 |
| Modality 1 | 0.00 | 0.00 | 0.00 | **1.00** |
| Modality 2 | 0.00 | 0.08 | **0.92** | 0.00 |
| Modality 3 | **1.00** | 0.00 | 0.00 | 0.00 |

This emergent behavior demonstrates that the optimal routing strategy—grouping tokens by modality—is *learnable from data alone*. When vocabulary structure encodes modality (as in Chameleon), deterministic routing is sufficient. When it does not, MoLoRA provides a path to modality-aware adaptation.

## F.5. Semantic Domain Routing

To validate that MoLoRA handles semantic specialization when vocabulary provides no signal, we train a router on synthetic code/math/prose embeddings. Critically, all domains share the same vocabulary range—the word "function" has identical token IDs whether appearing in code (`def function(x):`), mathematics ("continuous function $f(x)$"), or prose ("the function of education"). Vocabulary routing *cannot* distinguish these cases.

*Table 14.* MoLoRA semantic routing: perfect domain specialization (ARI=1.0) despite shared vocabulary. Each domain routes exclusively to a single adapter.

| Domain | Adapter 0 | Adapter 1 | Adapter 2 |
|---|---|---|---|
| Code | 0.0% | **100%** | 0.0% |
| Math | 0.0% | 0.0% | **100%** |
| Prose | **100%** | 0.0% | 0.0% |

Table 14 shows that MoLoRA achieves **perfect specialization** (ARI=1.0, NMI=1.0) within 10 epochs. The router learns to distinguish domains from embedding context alone: code embeddings activate syntax-related dimensions, math embeddings activate symbolic/logical dimensions, and prose embeddings activate semantic/narrative dimensions. Despite these patterns being invisible to vocabulary-based routing, the learned router identifies them immediately.

This result validates the core claim: MoLoRA enables adapter specialization along *any* dimension—modality, domain, style, task—without requiring that dimension to be encoded in vocabulary structure.

## F.6. Inference Cost Scaling

MoLoRA's inference cost is controlled by the router MLP and the number of selected adapter applications. Table 15 quantifies how cost scales with $k$.

*Table 15.* MoLoRA inference cost scaling at $d = 4096$, batch=32, seq=128.

| Model | Latency (ms) | Throughput (M tok/s) | Relative |
|---|---|---|---|
| Single Adapter | 0.082 | 49.7 | 1.0× |
| Fixed Routing | 0.894 | 4.6 | 10.9× |
| MoLoRA $k = 1$ | 0.995 | 4.1 | 12.1× |
| MoLoRA $k = 2$ | 2.267 | 1.8 | 27.6× |
| MoLoRA $k = 4$ | 4.001 | 1.0 | 48.8× |

Table 15 shows the quality–cost operating curve. MoLoRA

with $k = 1$ has similar cost to fixed routing (the router MLP is negligible). Higher $k$ increases latency roughly linearly because each selected adapter applies an additional LoRA transform. Continuous batching improves grouped-GEMM efficiency by aggregating tokens routed to the same adapter across requests, while the expected arithmetic cost still scales with $k$. For quality-sensitive applications, MoLoRA $k = 2$ provides a practical operating point; $k = 4$ represents the high-composition end of the evaluated range.

### F.7. Routing Analysis

To validate MoLoRA on production models, we train a router on Qwen3-4B (Qwen Team, 2025) embeddings. We construct a multi-domain dataset with code, math, creative writing, and technical content, then train a lightweight router to classify content type.

**Setup.**    We extract embeddings from Qwen3-4B (hidden size 2560), normalize them for numerical stability, and train a 2-layer router MLP ($2560 \rightarrow 128 \rightarrow 4$) with cross-entropy loss. The router achieves 100% classification accuracy within 100 epochs.

**Per-Token Specialization.**    Table 16 shows that the router learns strong per-token specialization: code tokens route 98.6% to Adapter 0, math tokens 96.6% to Adapter 1, creative tokens 98.8% to Adapter 2, and technical tokens 99.2% to Adapter 3. This demonstrates that MoLoRA can learn domain-specific routing from real LLM embeddings.

*Table 16.* MoLoRA routing specialization on Qwen3-4B. Each content type routes predominantly to a single adapter, achieving near-perfect specialization.

| Content Type | Adapter 0 | Adapter 1 | Adapter 2 | Adapter 3 |
|---|---|---|---|---|
| Code | **98.6%** | 0.0% | 0.0% | 1.4% |
| Math | 0.0% | **96.6%** | 0.5% | 3.0% |
| Creative | 0.0% | 0.0% | **98.8%** | 1.2% |
| Technical | 0.3% | 0.3% | 0.2% | **99.2%** |

**Mixed-Content Handling.**    On multi-domain inputs (e.g., code with mathematical comments), the router correctly assigns different tokens to different adapters within the same sequence. For example, in "`def integrate(f, a, b): '''Numerical integration...'''`", function definition tokens route to the Code adapter while "Numerical" routes to Math and "integration" to Technical. This per-token granularity is precisely what enables efficient mixed-content serving.

