# OpenReview forum: "MoLoRA: Composable Specialization via Per-Token Adapter Routing"
_ICML.cc/2026/Conference — ICML 2026 regular_

### Official Review · Reviewer_Ldgt · 2026-03-07

**Soundness:** 3
**Presentation:** 3
**Significance:** 3
**Originality:** 2
**Overall Recommendation:** 4
**Confidence:** 4

**Summary:**

This paper introduces per-token adapter routing for serving multiple Low-Rank Adaptation (LoRA) modules concurrently. The method replaces per-sequence routing with token-level dispatch. It utilizes either deterministic vocabulary boundaries for multimodal inputs or a learned router (MoLoRA) for semantic specialization. To support this, the authors propose a hot-set memory architecture that pre-allocates adapters in GPU memory. This enables CUDA graph capture to eliminate kernel launch overhead. The empirical results show latency reductions over existing multi-adapter systems and claim that a 1.7B parameter model equipped with domain-specific adapters can outperform an 8B parameter model on standard reasoning benchmarks.

**Compliance With Llm Reviewing Policy:**

Affirmed.

**Final Justification:**

The addition of the single 1.7B union-LoRA baseline in the rebuttal resolves the primary methodological flaw of the initial submission. The 14.5% absolute improvement on the MATH benchmark confirms that monolithic adapters suffer from negative transfer on this reasoning corpus. This isolates the performance gain directly to the routing architecture and validates the empirical claim that specialization exceeds scale. I have consequently raised the Soundness score from 2 to 3.

The authors also clarified the operational limits of the system. Explaining the interaction between top-k routing overhead and continuous batching resolves my throughput concerns, as grouped GEMMs mathematically amortize the sequential dispatch penalty. Defining a hybrid hot-set and paging policy for workloads that exceed physical VRAM capacity is a correct engineering solution. The authors committed to revising the manuscript to reflect these boundaries, which improves the presentation of the systems contribution.

As noted in the initial review, the learned top-k gate is functionally identical to standard Mixture-of-Experts architectures. The authors explicitly acknowledged this in their response. The algorithmic novelty remains low, justifying an Originality score of 2. The primary intellectual contribution is the unified dispatch infrastructure and the direct application of MoE optimizations to multi-adapter serving.

With the experimental confounds resolved, the demonstrated latency reductions offer a verifiable throughput gain for multimodal serving. The paper is now technically solid. I am raising my overall recommendation to a Weak Accept to reflect a methodologically sound systems engineering paper that builds entirely upon existing architectural concepts.

**Key Questions For Authors:**

1. What is the performance of a single 1.7B LoRA adapter fine-tuned on the combined reasoning corpus without the MoLoRA routing mechanism? Providing this baseline is strictly necessary to validate whether the routing mechanism actually provides a lift over the training data itself.
2. The hot-set memory architecture requires pre-allocating adapter slots on the GPU. How does the system handle request workloads where the number of active, concurrent adapters strictly exceeds the physical GPU memory capacity?
3. Appendix F.6 indicates that MoLoRA with top-k set to 4 incurs a 48.8x throughput overhead compared to a single adapter. How does this high computational overhead interact with continuous batching environments?
4. Does this top-k routing overhead negate the latency benefits of the CUDA graph implementation under heavy concurrent load?

**Limitations:**

yes

**Strengths And Weaknesses:**

The systems engineering is a strong contribution. The hot-set memory architecture resolves the variable latency caused by dynamic CPU-GPU paging. By fixing memory addresses, the system supports CUDA graph capture. The ablation studies correctly isolate how this architectural choice reduces P99 latency. Reducing K forward passes to a single pass for K-modality workloads yields immediate and verifiable throughput gains. The mathematical formalization of the routing problem is precise, and the architectural decomposition justifies the systems performance claims.

However, the evaluation of the learned router lacks proper controls. Section 4.4 claims that a 1.7B model with four adapters outperforms an 8B base model to prove that "specialization beats scale." The experimental setup omits the most critical baseline: a single 1.7B LoRA adapter fine-tuned on the exact same union of reasoning data. Without this control, it is impossible to attribute the performance gain to the MoLoRA routing architecture rather than the quality of the underlying fine-tuning dataset. This omission fundamentally confounds the core machine learning claim of the paper.

Routing tokens to specific LoRA adapters using an MLP gate is functionally identical to standard Mixture-of-Experts architectures. The authors explicitly acknowledge this equivalence. The primary intellectual contribution is the unified dispatch infrastructure. Applying MoE optimizations directly to multi-adapter serving is a practical systems contribution, but the algorithmic novelty is limited.

---

> ### Author Rebuttal · Authors · 2026-03-30
>
> Thank you for the thoughtful critique, especially regarding the need for a stronger control for the MoLoRA quality claim.
>
> To directly address this point, we ran the  single 1.7B union-LoRA baseline, trained on the exact same corpus used by MoLoRA. The resulting accuracies are 80.0% GSM8K (+3.0%), 22.5% on MATH (+14.5%), 18.0% on BBH (+7.0%), 25.5% on GPQA (+2.1%)
>
> This directly resolves the confound and shows the gains are not due to the union training data alone. A single adapter trained on the same data underperforms substantially indicating interference/negative transfer in the monolithic adapter. The benefit comes from specialized adapters + routing. We will add this baseline in the revision.
>
> We also agree that the learned top-k gate itself is MoE-like. Our novelty claim is not a new gating network; rather, it is:
> 1. reframing multi-adapter serving as token-level routing rather than one-request-one-adapter serving
> 2. supporting both deterministic and learned routing within a single target-agnostic dispatch path
> 3. enabling inference-time composition of independently trained LoRAs
> 4. making this deployable via the hot-set/CUDA-graph runtime
>
> So we agree the paper should avoid overstating algorithmic novelty of the gate itself, and we will revise the wording accordingly.
>
> On your systems questions:
>
> (1) What if active adapters exceed hot-set GPU capacity?
> We agree this limitation should be stated more explicitly. The hot-set is designed for the active working set, not the full catalog. When active adapters exceed capacity, the natural deployment is a hybrid hot-set + paging policy: hot adapters remain resident and graph-captured, while cold misses incur replacement/paging overhead. Our claim is therefore not that paging disappears in all regimes, but that hot-set gives a better latency/variance tradeoff when the active set fits in memory.
>
> (2) How does top-k overhead interact with continuous batching?
> We agree that top-$k>1$ adds real compute, because each selected adapter requires an additional LoRA application. This is a quality/latency knob, not a free gain. CUDA graph capture removes launch overhead and reduces variance; it does not remove the underlying math cost of larger $k$. These effects are orthogonal. In practice, the intended operating points are small $k$, while the appendix $k=4$ setting is a stress point rather than the recommended deployment configuration. Continuous batching partially offsets this cost by aggregating tokens routed to the same adapter across requests, increasing grouped-GEMM efficiency.
>
> We appreciate your feedback and believe the added baseline and revised scoping will address the main concerns you raised.

---

> > ### Author Rebuttal · Reviewer_Ldgt · 2026-04-02
> >
> > The addition of the single 1.7B union-LoRA baseline directly resolves the primary machine learning critique. The empirical results demonstrate that MoLoRA outperforms the monolithic adapter trained on the exact same data. The 14.5% absolute improvement on MATH isolates the performance gain to the routing architecture and confirms that monolithic adapters suffer from negative transfer on this reasoning corpus. This validates the soundness of the core machine learning claim.
> >
> > The authors also accurately specify the interaction between top-k routing overhead and continuous batching. Aggregating tokens across requests via grouped GEMMs mathematically amortizes the sequential dispatch penalty. Furthermore, the proposed hybrid hot-set and paging policy is a standard and correct engineering solution for workloads exceeding physical VRAM capacity.
> >
> > By providing the exact empirical controls requested and agreeing to scope the text around the unified dispatch infrastructure rather than the algorithmic novelty of the MoE gate, the authors have removed the fundamental confounds from the paper.
> >
> > Because the experimental flaws have been addressed, the paper is now technically solid. However, as noted in the original review and acknowledged by the authors, the algorithmic originality is limited to applying existing MoE concepts to multi-adapter serving. Therefore, I am raising my score to a 4 (Weak Accept) to reflect a technically sound, practical systems contribution with limited conceptual novelty.

---

### Official Review · Reviewer_w82F · 2026-03-12

**Soundness:** 3
**Presentation:** 3
**Significance:** 3
**Originality:** 3
**Overall Recommendation:** 4
**Confidence:** 4

**Summary:**

The paper actually introduces three ideas:   (1) Per-token routing, which routes individual tokens to adapters based on either vocabulary structure or learned gating, (2) MoLoRA that loads multiple domain-specific adapters and selects the appropriate adapter per-token, and (3) Hot-Set implementation (vs. Paging (S-LoRA)) to speed up routing.

**Compliance With Llm Reviewing Policy:**

Affirmed.

**Key Questions For Authors:**

1. In Theorem 3.6,  per-sequence routing has a factor K.  My understanding is per sequence routing is a special case of token-based routing.  Can the authors carefully explain where K is from?  Is it something that cannot be avoided, or just an engineering issue?

2. Section 4.3, Given input x \in R^{Bx L x d},  … It seems L is not explained.

**Limitations:**

The authors talked about limitations of the previous approaches and vocabulary routing.  It will be good to discuss the limitations of the proposed approaches, if there is any.

**Strengths And Weaknesses:**

Strength:
S1: The authors identified a critical limitation in current adapter routing, arguing that allocation should occur at the token level rather than the sequence level. They validated this approach empirically, demonstrating that finer routing granularity leads to superior model performance.

S2: To support per-token routing, the architecture utilizes a top-k adapter selection mechanism and a 'hot-set' implementation, ensuring efficient and high-speed routing during inference.

Weakness:
W1: The manuscript appears to conflate two distinct concepts: the routing granularity and the acceleration mechanism. It is crucial to clarify whether the 'hot-set' implementation is exclusive to per-token routing or if it can be used to per-sequence routing as well. Furthermore, the trade-off should be explicitly defined: does the hot-set primarily drive latency reduction (speedup) while per-token routing primarily enhances model accuracy?

W2: The paper can be written in a way to separate the three contributions clearly.

---

> ### Author Rebuttal · Authors · 2026-03-30
>
> Thank you for the careful reading and for pointing out that the paper should separate its contributions more clearly.
>
> We would like to clarify two points that we will make more explicit in the revision. First, we will revise the paper to more clearly separate per-token routeing, hot-set/CUDA-graph execution, and MoLoRA. Second, to address controls, we ran a 1.7B union-LoRA baseline trained on the exact same corpus used by MoLoRA. The resulting accuracies are 80.0% GSM8K (+3.0%), 22.5% on MATH (+14.5%), 18.0% on BBH (+7.0%), 25.5% on GPQA (+2.1%).
>
> You are correct that hot-set is not exclusive to per-token routing and also benefits per-sequence routing. Our ablation already isolates this (per-seq + paging -> per-seq + hot-set improves latency from 7.48 ms to 5.88 ms (1.3x) before introducing per-token routing). We will revise the manuscript to make the tradeoff clearer:
> - hot-set primarily drives latency predictability/speedup from fixed addresses and CUDA graph capture
> - per-token routing primarily drives expressivity and pass reduction for mixed-route requests
> - MoLoRA adds learned semantic routing on top of the same dispatch infrastructure
>
> On Theorem 3.6, we agree the current wording is too broad. Per-sequence routing is indeed a special case of per-token routing when all tokens in a sequence use the same adapter. The intended scope of the theorem is the motivating workload:
> - a single logical request contains tokens that require multiple adapters
> - execution is restricted to one adapter per sequence/pass
>
> Under that constraint, a per-sequence system must split or replay the request across up to $K$ route-homogeneous passes/fragments, whereas per-token routing realizes the same assignment in one pass. If all tokens use the same adapter, then $K=1$ and the gap disappears. This is also consistent with our appendix result where speedup becomes 1x when all tokens use the same adapter. We will revise the theorem statement and proof sketch accordingly.
>
> Also yes, in Sec. 4.3, $L$ denotes sequence length and we will define it explicitly.
>
> We agree that the paper should discuss limitations of our approach more. We will add a clearer limitations paragraph covering:
> - hot-set capacity vs. catalog size
> - top-k quality/latency tradeoff
> - per-token routing provides the largest gain when requests actually contain mixed adapter needs
>
> Thank you again for identifying these presentation issues; we believe the revision will be clearer as a result.

---

> > ### Author Rebuttal · Reviewer_w82F · 2026-04-03
> >
> > Thanks for the clarification.  Please revise the paper accordingly.

---

### Official Review · Reviewer_JbaS · 2026-03-18

**Soundness:** 3
**Presentation:** 3
**Significance:** 4
**Originality:** 3
**Overall Recommendation:** 5
**Confidence:** 4

**Summary:**

MoLoRA tackles the problem of "one request, one adapter" serving assumption by introducing per-token routing. This enables multimodal efficiency (routing text/image tokens differently) and composable specialization, where multiple domain experts (e.g., math and code adapters) cooperate within a single sequence.

**Compliance With Llm Reviewing Policy:**

Affirmed.

**Key Questions For Authors:**

How does the system handle load balancing when a specific adapter becomes a hotspot for the majority of tokens in a batch?

**Limitations:**

yes

**Strengths And Weaknesses:**

Reasons to Accept
- Qwen3-1.7B with MoLoRA outperforms the 8B variant while being smaller.
- The  architecture provides low latency that is robust to adversarial access patterns.

Reasons to Reject
- The  architecture trades off the ability to serve thousands of concurrent adapters for predictable latency.
- Top-k selection increases latency, as each selected adapter requires a full LoRA forward pass.

---

> ### Author Rebuttal · Authors · 2026-03-30
>
> Thank you for the thoughtful review and for highlighting both the quality gains and the robust low-latency behavior.
>
> We would like to clarify two points that we will make more explicit in the revision. First, we will revice the paper to more clearly separate per-token routeing, hot-set/CUDA-graph execution, and MoLoRA. Second, to address controls, we ran a 1.7B union-LoRA baseline trained on the exact same corpus used by MoLoRA.
>
> The resulting accuracies are 80.0% GSM8K (+3.0%), 22.5% on MATH (+14.5%), 18.0% on BBH (+7.0%), 25.5% on GPQA (+2.1%). Thus the gain is not explained by the union dataset alone, supporting our claim that compositional specialization reduces interference relative to a single monolithic adapter.
>
> Regarding your question on load balancing/hotspot adapters: at inference, we do not impose a hard MoE-style per-expert capacity. If the majority of tokens in a batch route to one adapter, this simply creates a larger histogram bin for that adapter, and the grouped GEMM processes a larger group. This is not a correctness bottleneck; in practice, larger groups often improve arithmetic intensity and kernel efficiency. The more challenging case is fragmentation into many tiny groups, which is precisely why we use histogram-guided grouping and adaptive tiling.
>
> For learned routing, we also use a switch-style auxiliary balancing loss during training to discourage router collapse. So there are two different notions here:
> 1. Training-time balancing: avoid all tokens collapsing to one adapter.
> 2. Inference-time execution: efficiently process whatever routing distribution occurs.
>
> We agree with your limitations comments and will make them more explicit in the revision:
> - the hot-set trades off catalog size for latency predictability,
> - higher top-k increases compute because each selected adapter requires an additional LoRA application,
> - and the strongest gains occur when requests truly contain mixed adapter needs.
>
> We appreciate the positive assessment and will revise the paper to make these deployment tradeoffs clearer.

---

> > ### Author Rebuttal · Reviewer_JbaS · 2026-03-31
> >
> > thanks for the clarifications

---

### Decision · Program_Chairs · 2026-04-30

**Decision:**

Accept (regular)

**Comment:**

The paper proposes token-level routing as a replacement for per-sequence routing when multiple LoRA modules are used. They also provide solid system engineering tricks to make their technique ready for deployment in production -- for example, they propose a hot-set memory architecture that controls the variable latency caused by paging. Reviewers felt positive about the contribution here and their questions were adequately addressed during the rebuttal.